# Evaluation of Total Mercury in Sediments of the Descoberto River Environmental Protection Area—Brazil

**DOI:** 10.3390/ijerph17010154

**Published:** 2019-12-24

**Authors:** Joelma Ferreira Portela, João Pedro Rudrigues de Souza, Myller de Sousa Tonhá, José Vicente Elias Bernardi, Jérémie Garnier, Jurandir Rodrigues SouzaDe

**Affiliations:** 1Analytical and Environmental Chemistry Laboratory, Instituto de Química, University of Brasilia, Brasilia, Federal District 70919-970, Brazil; joaoprsouza@outlook.com (J.P.R.d.S.); rodsouza@unb.br (J.R.S.); 2Geochemistry Laboratory, Institute of Geosciences, University of Brasilia, Brasilia, Federal District 70919-970, Brazil; myllerquimico@gmail.com (M.d.S.T.); garnier@unb.br (J.G.); 3Life and Earth Sciences Laboratory, University of Brasilia, Planaltina, Federal District 73345-010, Brazil; bernardi.jve@gmail.com

**Keywords:** accumulation, water reservoir, total mercury, sediment, descoberto river basin

## Abstract

To evaluate the total mercury accumulation (THg) in the Descoberto river basin environmental protection area (DREPA), nine sediment and water samples were collected from the Descoberto reservoir (lentic environment), and 23 in its tributaries (lotic environment), which are located in a densely urbanized area within the Descoberto river watershed, Brazil. The following physicochemical parameters of water were determined: dissolved oxygen (DO); hydrogen potential (pH); total dissolved solids (TDS); nitrate (NO_3_^−^); chloride (Cl^−^); temperature (T); sulfate (SO_4_^2−^), and in sediment, the concentration of total mercury (THg) and volatile material (VM) was determined. THg concentrations in sediments showed a significant difference (*p* = 0.002) between tributaries (0.03 µg g^−1^ ± 0.02) and reservoir (0.08 µg g^−1^ ± 0.04), indicating accumulation in the lentic environment. Most of the results evaluated for ecotoxicological risks presented values below the concentration, at which adverse effects would rarely be observed, ERL (effects range low). However, in relation to the enrichment factor (EF), applied to identify the anthropogenic contribution, the results indicate that most of the samples are moderately polluted through atmospheric deposition due to vehicular traffic and agriculture. These results show that the likelihood of methylation in the lentic environment is higher than in the lotic environment.

## 1. Introduction

Sediments are of great interest when evaluating contamination in water reservoirs, which are often located near anthropized areas and are spotlights for several impacts [1]. In these systems, there is evidence of contaminants in water as well as in sediments, due to the matter transfer mechanisms between these two phases [2]. Chemical, physical, and biological processes can lead to the release of contaminants in the sediments to the water column [3]. In this sense, surface sediments, which reflect the current state of contamination, are considered a great means to assess environmental quality [4].

Among the hazardous substances entering these environments, mercury (Hg), a global pollutant, has attracted the attention of environmental scientists since the deaths of thousands of people in Minamata, Japan in the 1950s after exposure to methylmercury [5]. In 2017, the Minamata Convention came into force with the aim of reducing anthropogenic mercury (Hg) emissions worldwide. Monitoring of Hg species is required in order to effectively reduce exposure in humans [6]. In aquatic systems, Hg is present in its elemental, inorganic, and organic forms, and its ionic forms can be converted to the highly toxic monomethylated species (HgCH_3_^+^) by iron- and sulfate-reducing bacteria. Sediments act as an important site for Hg transformations and a source of methylmercury which can be bioaccumulated and biomagnified in the food chain [7,8].

Mercury transport in aquatic systems is influenced by environmental factors, and several environmental variables show a correlation between total mercury (THg) in sediments with organic matter content, chloride concentration, temperature, among others [9]. The flooding of areas previously covered with vegetation, like the case of reservoirs for public water supply, changes the physicochemical characteristics of the site and creates conditions for the conversion of Hg to its most toxic species. In the Amazon rainforest, for example, which is a complex aquatic environment, the biological and physicochemical properties of its water bodies are linked to its biogeochemical origin and the drainage of watersheds. Moreover, these properties have a detectable influence on the distribution and availability of Hg species [10].

Also, the disturbance of Hg cycling in central Amazon soils caused by deforestation and forest exploitation results in increased Hg contamination in the lower Tapajós lentic systems. Exports of mercury containing fine particulate matter from the soil surface may be associated with iron and aluminum oxyhydroxides, which facilitates their transport to lake systems [11].

The terrestrial environment represents the largest inventory of Hg (approximately 950 kt), with approximately 150 kt stored in surface organic soils. The atmospheric and aquatic chemistry of Hg is one of the most complexes of all trace metals [6]. Metallic mercury slowly volatilizes at room temperature [12] and its vapor is stable in the atmosphere, thus it can be transported on a global scale through air currents, affecting remote natural areas away from point sources of contamination. Mercury is present in various raw materials such as coal, oil, wood, and various mineral deposits, and can be released into the air or other means when these materials are burned, processed, or disposed of. Among human activities, the burning of fossil fuels is the most important in terms of volume and distribution [13].

Diffuse pollution, including agriculture, deposition of air pollutants, and surface runoff, is a complex pollution type and has strong links with eutrophication, so it is recurrently the object of research. River basin pollutants can be transferred from land to water and accumulate in sediments [14]. In the case of reservoirs, flooded soils and vegetation may also release Hg to the water column after the flooding [15]. These changes in the landscape and land use in the watershed also interfere in the organic carbon flow and redox conditions that directly influence the Hg methylation processes in adjacent aquatic systems. The Hg concentration in reservoir fish are mainly controlled by the influx and dynamics of organic carbon and terrestrial Hg even decades after reservoir damming [6].

Thus, studies that consider the characteristics of each region are necessary for contributing to land use management and avoiding the deterioration of aquatic ecosystems [16]. Evaluation of mercury levels in tropical reservoirs without direct influence from known anthropogenic sources has not been frequently conducted [17], particularly in the Brazilian Cerrado. This study aimed to evaluate the distribution of THg levels in the bottom sediments of the Descoberto reservoir (Federal District, Brazil), built in 1979 to supply potable water to the Federal District, and in tributaries of the Descoberto river watershed, including the Descoberto river environmental protection area (DREPA). This region suffers from the intensification of diffuse pollution, as showed in a geochemical study of the stream sediment of the Descoberto river watershed (DRW) carried out in 2005, in which the concentrations of P, Mn, Ba, Ca, Mg, and K demonstrated the effects of anthropic activity in this region. However, studies in the DREPA on the indispensable comparison between Descoberto Lake and its tributaries regarding mercury accumulation are lacking.

Given this, the objectives of this research were: (1) to evaluate the distribution of Hg considering two different environments, lotic (tributaries) and lentic (reservoir); (2) to evaluate the environmental threats and the anthropogenic enrichment of this metal based on the sediment quality guidelines (SQG) and enrichment factor (EF); and (3) to analyze the relationships between the amount of THg, the tributaries, and the reservoir, while also considering the mineralogical composition of the sediments.

## 2. Materials and Methods

### 2.1. Study Area

The study area corresponds to 41,064 hectares and is located in the Descoberto river basin between 15,711° S and 48,152° W (Figure 1). The DREPA is contained in the Cerrado biome, and it is responsible for about 66% of the public water supply of the Federal District, while also being an important agricultural hub. With this, the region has presented agricultural and urban expansion to the detriment of areas of natural vegetation over the years. The growth of these activities over the years was carried out in a disorderly manner, with irregular soil occupation and deforestation, compromising the quality of water resources [18]. The Paranoá group is the only geological unit that outcrops in the Descoberto river environment protection area (DREPA), and among the lithostratigraphic units of this group are the following geological assemblies: slate unit (MNPpa); sandy metarhythmite unit (MNPpr3); average quartzite unit (MNPpq3); and clay metarhythmite unit (MNPpr4) [19].

We sampled 32 points in the reservoir and in the tributaries intercepted by highways (Figure 1). Sampling was made up and downstream of the roads. It was carried out in the tributaries in May 2017, and in the reservoir in May 2018, at the time of transition from dry to the rainy season. Sediment sampling occurred in rivers Descoberto (P1, P2, P15, P16, P21, P22), Rodeador (P9, P10, P11, P12, P13, P14), Rio das Pedras (P17, P18, P19, P20), in the streams Caboquinho (P3, P4), Chapadinha (P5, P6), Olaria (P7, P8), and Capão da Onça (P25, P26), and in the reservoir (L1, L2, L3, L4, L5, L7, L8, L9, L10).

### 2.2. Sampling

#### Sediment and Water

Sediment samples in the tributaries were collected from the riverbed with a plastic shovel. In the reservoir, a gravity tubular sampler of the kajak type was used. The sediment samples were packed in polyethylene bags and approximately 1 kg of each point was collected. Surface water in the tributaries and reservoir was collected approximately 20 cm from the surface in 1 L polyethylene vials, which were previously left in HCl 10% solution for at least 8 h and rinsed with water from the Milli-Q purification system. When collecting water samples, the vials were rinsed with water from the sampling point before sample storage. All samples were identified, transported to the laboratory, and kept refrigerated (4 °C) until sample preparation.

### 2.3. Sediments and Water Samples Preparation

The sample preparation was performed at the UnB geochemistry laboratory. Sediment samples were dried at room temperature, disaggregated, and sieved to a fraction smaller than 63 µm. This fraction was used for volatile material (VM) determination by loss on ignition (in muffle furnace at 500 °C), mineralogical analysis, and aluminum determination (Al), used to calculate the background value of the study area.

For anion determination in water, the samples were filtered using *Milipore*^®^ cellulose esters membranes of 0.45 µm pore size.

#### 2.3.1. Determination of Mercury and Aluminum in Sediment

The determination of total mercury (THg) was performed at the laboratory of analytical and environmental chemistry at UnB. Wet sediments were utilized, and sample analyses were performed in triplicate, where the coefficient of variation between repetitions was 0.5 to 21.0%, San Joaquin soil (NIST 2709) was used as reference material, with a 120% recovery (*n* = 6). The limit of quantification was 0.0015 µg g^−1^. Quantification was performed by atomic absorption spectrophotometry on a Lumex Instruments RA915+ with Zeeman correction, in which sample decomposition occurred on a first chamber heated up to 740 °C, and analyte atomization took place on a second heated chamber heated up to 700 °C. Ambient air was the carrier gas that passed through both chambers at 1.0 ± 0.2 L min^−1^ air flow rate. Samples were weighed in a high precision balance and directly inserted into the first heated chamber on a quartz boat.

Aluminum (Al) determination was performed at the UnB geochemistry laboratory. Extraction of the metal contained in the pellet was carried out by fusion with lithium metaborate (LiBO_2_) at 900 °C in platinum crucibles and further digestion in hydrochloric acid (HCl). Quantification was performed by ICP-OES (5100, Agilent, Hainesport, NJ, USA), accuracy was determined using validated standard materials (BHVO and NIST 1646a).

#### 2.3.2. Water Physicochemical Parameters Determination

Using the HORIBA model U-52G multiparameter probe in the field, temperature, pH, electrical conductivity (EC), turbidity (tur), dissolved oxygen (DO), and total dissolved solids (TDS) were determined. Alkalinity was determined by tiltulometry with standardized 0.001 mol L^−1^ H_2_SO_4_. Anions concentrations (chloride, nitrate, and sulfate) were determined at the geochemistry laboratory by suppressed conductivity ion chromatography (IC) in a Dionex equipment model ICS90, with an ion-exchange column with functionalized surface of quaternary alkylammonium, using Na_2_CO_3_/NaHCO_3_ as eluent, 1.2 mL min^−1^ flow, 1500 psi pressure, and injection volume equal to 25 µL. The analytical methodology for water analysis was applied to all samples following the criteria and standards of the standard methods for the examination of water and waste water, of the American public health association (APHA), American water works association (AWWA), and the water pollution control federation (WPCF) [20].

### 2.4. Sediment Mineralogy

The mineralogical composition of the sediment was determined on powdered samples in an X-ray diffractometer (XRD; Rigaku^®^, Ultima IV) using Cu-KαNi-filtered radiation and monochromator graphite at the Mineralogical Laboratory of UnB. The XRD patterns were recorded from 2° to 80° 2θ with a scan speed of 2° min^−1^. This was performed for the tributary’s samples downstream of the highways and all reservoir samples.

### 2.5. Statistical Analysis

To evaluate statistical differences between the lotic (tributaries) and lentic (reservoir) environments, considering as physical–chemical characteristics of water, the T-test was utilized for variables with normal distribution and the Mann–Whitney test (U) for variables with non-normal distribution. Determination of variables normality was performed through the Kolgomorov–Smirnov test.

For comparing the amount of THg between the aquatic environments, and to determine the influences (water body and road) that ranks in the most efficient way the accumulation of Hg in the environment, multivariate analysis was used. Hierarchical cluster analysis was performed by complete linkage, or farthest neighbor method, and dissimilarity correlation (euclidean distance) were utilized. Additionally, the cophenetic correlation coefficient (ccc) was calculated to evaluate the degree of deformation caused by the construction of dendrograms. For this, the XLSTAT software (Addinsoft, Boston, MA, USA) was used.

### 2.6. Risk and Pollution Assessment by Geochemical Indices and Sediment Quality Guidelinas 

The extent of contamination and the likelihood of environmental impact due to the presence of THg in DREPA were assessed using geochemical indices. For this, we used the Al-normalized enrichment factor index (EF) and consensus-based sediment quality guidelines (SQGs), which predict whether an adverse effect may occur or not [21,22].

The enrichment factor is an index that allows enrichment evaluation of an element by normalization with another element considered more stable and immovable in the environment for the purpose of evaluating the anthropic input of an element. Equation (1) was applied to calculate the EF of the sampling sites:EF_Hg_ = (THg/Al)_sample_/(THg/Al)_ref_(1)
where (THg/Al)_sample_ and (THg/Al)_ref_ are the concentration ratios in the sediment sample and the reference sample (background), respectively. When EF values are below 2, they do not indicate anthropogenic contamination, while values between 2 and 5 represent moderate contamination, and environmental pollution is considered significant with values between 5 and 20. Between 20 and 40, it is considered a high level of pollution [23].

The sediment quality guidelines (SQGs) are designed to establish criteria for assessing sediment quality and the toxicological significance of sediment-associated substances to aquatic organisms. Regarding the assessment of environmental threats resulting from toxic metals in sediments, two SQGs ranges were adopted, one referring to concentrations below, for which adverse effects on sediment fauna will be rare (ERL), and the other representing concentrations above which negative influences on aquatic system organisms are likely to occur (ERM) [24].

## 3. Results and Discussion

### 3.1. Difference between Lotic and Lentic Environment

The results of THg and the volatile material in sediment, as well as physicochemical parameters in DREPA water, are presented in Table 1. In general, the highest values of the variables were found in the lentic environment, except for DO, Turbidity (Tur), HCO_3_^−^, NO_3_^−^, and VM. There were significant variations (*p* ˂ 0.05) between lentic and lotic environments in DO, T °C, Cl^−^, NO_3_^−^, SO_4_^2−^ and THg, but not in pH, EC, Tur, TDS, HCO_3_^−^, and VM.

The pH values in the sampled sites showed no significant difference (*p* = 0.983) between the lentic (6.81) and lotic (6.83) systems, being within the average of the rivers of the Federal District (5.6 to 6.85) [25]. The pH is an important variable to evaluate Hg behavior in the aquatic environment, because, among other factors, it can favor mobilization and methylation processes in acidic waters, or hinder it in alkaline environments [10].

The mean DO value in the tributaries was 26.35 mg L^−1^, and in the reservoir it was 15.16 mg L^−1^. The solubility of dissolved oxygen is mainly affected by water temperature, which presented a higher mean value in the reservoir (22.30 °C) than in the tributaries (19.22 °C). Other environmental factors that may influence the amount of dissolved oxygen in lentic systems are organic pollution level, wildlife distribution densities, aquatic plants, and precipitation patterns [26].

Nitrate has been used as an indicator of eutrophication in reservoirs and has its origin in the use of agricultural fertilizers and animal breeding. Nitrate contents in the lotic environment (0.80 mg L^−1^) were higher than in the lentic environment (0.63 mg L^−1^), and that may be explained considering that effluents with high nitrate concentrations have their contents diluted when reaching the waters of the rivers [25], which would explain the decrease of this pollutant in the Descoberto reservoir.

Chloride levels showed an average value of 1.31 mg L^−1^ in the tributaries and 2.28 mg L^−1^ in the reservoir, which may indicate the discharge of domestic effluents in the DREPA, since the presence of chloride may be attributed to the contribution of raw sewage in tropical regions [27]. The data obtained showed an increase in the sulfate content in the reservoir (0.81 mg L^−1^), compared to the tributaries (0.46 mg L^−1^). This result may be associated with weathering and transportation of materials to the riverbed, but also with the discharge of domestic sewage [25].

The THg concentrations were higher in the lentic environment than in the lotic environment, and a comparison through the Mann–Whitney (U) test between the two systems showed a significant difference (*p* = 0.002) between tributaries (0.03 µg mg^−1^ ± 0.02) and reservoir (0.08 µg mg^−1^ ± 0.04), pointing to an accumulation of this metal in the lentic environment (Figure 2). Direct analysis of Hg in sediments is a reliable indicator of its distribution, since sediments represent the main environmental compartment for methylation of inorganic Hg. However, mobilization of sediment-bonded Hg depends on environmental conditions [10], making it important to evaluate the correlations of the physicochemical parameters of water with THg in order to verify possible processes of accumulation, transport, and transformation of this metal in the aquatic environment. Areas that house lotic and lentic ecosystems are governed by several factors, among them anthropogenic activities, hydrogeochemical processes, and local climatic conditions, these being the main influences of water quality conditions [26]. These environmental factors contribute to the accumulation or evasion of mercury in aquatic systems [28].

Table 2 shows a comparison between the amount of THg in sediment from other regions of Brazil and the Descoberto reservoir. The average in the Descoberto reservoir corresponds to the amounts of THg found in the Madeira River and the Rio Negro, in the northern region of the country, where several studies indicate the occurrence of mercury methylation in the sediment, due to the biogeochemical characteristics of these aquatic systems.

There is a significant potential for mercury methylation when an aquatic environment changes from lotic (flowing water) to lentic (low water flow). Methylation occurs preferentially in lentic system aquatic environments, with low oxygen concentration, acid pH, and availability of organic matter [29]. In the case of the Descoberto reservoir, the average for dissolved oxygen is 15.16 mg L^−1^, lower than the lotic environment (26.36 mg L^−1^), while there is an average pH of 6.8 for both environments. There was no statistical difference in volatile materials in lotic and lentic environments, which are linked to the presence of organic matter. Therefore, it appears that the Descoberto reservoir is acquiring characteristics favorable to the methylation of mercury.

The behavior of elemental mercury (Hg^0^) in aquatic systems depends on its chemical properties, but also on the properties of the environment in which it is inserted. This is because its chemical transformations can happen by photochemical, abiotic, and biotic reactions. Chemical transformations of Hg in aquatic systems comprise oxidation of Hg^0^, reduction of Hg^2+^, and methylation of Hg^2+^ [7]. In this context, another variable that may contribute to the difference in THg concentration in these environments is the strong interaction between this metal in inorganic form with organic matter and clay minerals present in the lake sediments [30], a factor associated with the dynamics of lentic environments and the low water flow.

Thus, it is suggested that the spatial characteristics of the Descoberto lake, its relative position in the river basin, the water catchment area (less than 64 hectares), plus land use (including traffic routes, agriculture, and domestic sewage discharge), corroborates with the accumulation of Hg in the bottom sediments from the lentic environment. However, despite the accumulation of THg in DREPA’s lentic environment, water acidity does not contribute to mercury bioaccumulation. Studies on the mechanism of this relationship indicate that more acidic lakes have higher MeHg concentrations, leading to greater availability in the food chain [31].

#### Sediment Risk Assessment in Relation to THg

Table 3 presents the descriptive statistics for the THg and Al concentrations in the sediments in the study area, the reference values of the sediment quality guidelines (SQGs) and the background values for other regions and DREPA.

The different geological characteristics associated with anthropogenic activities in DREPA make it difficult to establish a control area that can provide background values for the EF calculation. Thus, the background values (BG) for this work were defined as the average of four collection points (P9, P17, P25, and L7), located in areas with low anthropic activity [32].

The reference values were compared with those established for the US and specifically for Lake Coeur d’Alene [33] and to Paracatu River, Minas Gerais [32]. The BG values defined for this work agree with those utilized in studies from different regions of the world. In general, the THg concentration doepresents a difference between the maximum (0.1757 µg g^−1^) and the minimum (0.0146 µg g^−1^) concentrations by comparison with background values (0.0382 µg g^−1^), which may indicate the distribution of concentrations of this metal, considering the different areas in the sediment in the DREPA.

The sediment quality guidelines (SQGs) were used to assess the potential ecotoxicological risks associated with Hg contamination. Results show that for Hg, 3.12% of samples are within the ERL and ERM range (0.15 µg g^−1^ < Hg < 1.3 µg g^−1^), and 96.87% are below the ERL (Hg < 0.15 µg g^−1^). It can also be observed that most of the data evaluated for ecotoxicological risks have values below ERL. In terms of toxicity, ERL values represent the concentration below, for which adverse effects would rarely be observed; values between ERL and ERM mean that adverse effects would occasionally occur; and ERM values represent the concentration above, for which adverse effects would often occur [34]. Therefore, according to the studied aquatic environments, the sample L10 located inside the reservoir presents an intermediate risk, because its Hg value of 0.18 µg g^−1^ exceeds the ERL limit.

The enrichment factor, applied to identify the anthropogenic contribution [22], indicates that most of the samples analyzed have minimal or no contamination (EF < 2). However, there are moderately polluted samples with 2 < EF_Hg_ < 5 (P18, L2, L5, L9), and significantly polluted with 5 < EF_Hg_ < 20 (L10). Point P18 is a point located downstream of DF 450, which is a traffic lane near the city of Taguatinga, the second most populated region of the Federal District. Points L2, L5, L9, and L10 are located within the reservoir and receive runoff from agricultural areas. The results suggested that, knowing there is no history of direct human activity in DREPA, the moderately polluted samples may be associated with atmospheric exposure caused by vehicle traffic and leaching of pesticide and fertilizer constituents used by agriculture.

### 3.2. Dissimilarity Analysis

Figure 3 shows the dendrogram for the amounts of THg considering all sampled sites in DREPA. In order to ascertain whether the information on Hg accumulation could be related to the type of aquatic environments studied—lentic or lotic—an analysis of agglomerative hierarchical grouping was performed by the complete linkage method.

The representativeness of the actual distances between the original unmodified data through the graphical representation of the dendrogram was adequate (ccc = 0.934), because the closer the ccc is to 1.0, the better the representation [35]. Thus, it was found that the grouping based on THg concentrations obtained was appropriate and that there are four distinct classes considering the grouped sampled sites, as shown in Table 4.

The similarity between different aquatic environments may be related with local characteristics and amounts of THg in sediments. Objects in each cluster are similar to each other, but different from objects in other conglomerates. Thus, the central objects for class 1 is P17 (THg = 0.032 µg g^−1^), for class 2 is P18 (THg = 0.136 µg g^−1^), for class 3 is L4 (THg = 0.076 µg g^−1^), and for class 4 is L10 (THg = 0.176 µg g^−1^).

Based on Figure 3 and Table 4, it was noted that the groups formed regarding the THg concentrations correspond to the type of aquatic environment, lentic, and lotic. Group 1 (C1) is basically formed by the sampling sites in the lotic environment, except for the presence of the point L3. This group is composed of two subgroups, where L3, P10, and P22 correspond to sites with no apparent correlation with the terrestrial pathway or drainage receiving water body.

In addition, it was observed that class 3 contains all sampling points in the lentic environment, indicating that the homogeneity within the group expresses the distribution condition of THg inside the reservoir. Classes 2 and 4 correspond to points L10 and P18, the locations with the highest amounts of THg. In summary, the cluster analysis for the 32 sampling sites, based on the content of THg, revealed four distinct classes (Table 3). The similarity between the sites is clearly related to the type of aquatic environment, however, the difficulty in correlating the subgroups, particularly concerning the lotic environment (C1), evidences the complexity in determining the sources of Hg, possibly due to the diversity of anthropogenic activities that may impact DREPA, causing diffuse pollution.

### 3.3. Mineralogical Analysis of Sediments

Table 5 shows the mineral constituents of the DREPA sediment samples. Quartz appears as the main constituent in all samples, followed by kaolinite, illite, and gibbsite, as minor constituents. Although not in all samples, rutile, goethite, anatase, saponite, calcite, dolomite, and hematite also appear.

Studies of mineralogical and geochemical compositions of surface sediments allow a better understanding of the source and destination of terrestrial materials in the basin, as well as the factors that control these materials’ distribution and geochemistry of sediments [36]. In this context, clay minerals play an important role in the mobility of elements in the surface environment due to their sorting capacity [5]. At basic pH, mercury has a higher affinity for mineral fraction and at acid pH for organic matter, however there is still controversy for understanding the biogeochemical cycle of Hg in the environment [37]. In this study, the average pH of 6.8 indicates that there is no predominance of mercury in the organic or mineral fraction.

The predominance of quartz, kaolinite, gibbsite, and illite is in agreement with the geology of the studied region. The XRD analysis of surface sediments suggests spatial uniformity of lake surface sediments (Figure 4) and that the origin of mercury in the study area is not associated with the natural geology of the region. The presence of mercury in areas without direct anthropogenic influence may be associated with the erosion and leaching of the mercury-containing particles associated with iron oxides and hydroxides (hematite and goethite), processes that are favored by deforestation. These materials have been identified as natural sources of mercury for aquatic ecosystems [30].

The Descoberto reservoir extends over lateritic crust surfaces under tropical climate and savannah vegetation. In general, typically clayey sediments rich in plant debris predominate in it. Carbonate concretions (containing goethite, gibbsite, quartz, and kaolinite) occur frequently and may form the substrate in the lentic environment. The most important substrates for metal retention must have a high cation exchange capacity. Kaolinite (Al_2_SiO_5_(OH)_4_), which is a clay mineral commonly found in tropical soils, and is part of the Descoberto reservoir sediment samples, has a known low adsorption capacity [38], which may allow the dissolution of mercury ions in the water column. Thereby, the aluminosilicates of the region can act as a support for Hg [11], contributing to its accumulation in DREPA’s lentic environment, which demonstrates that agricultural activity and deforestation are factors that also contribute to the accumulation of THg in the sediments of the Discovery lake.

## 4. Conclusions

This study analyzed the occurrence and mobility of mercury in the Descoberto river basin environmental protection area, considering the physicochemical characteristics of the aquatic system. The investigation was carried out on 32 sampling points upstream and downstream of highways and in the Descoberto reservoir, which has highlighted a strong influence of urbanization on some water quality parameters, and the accumulation of Hg concentration in the reservoir sediment to public supply. Physicochemical parameters, such as slightly acidic pH, suggest that the waters do not easily extract mercury in the sediments, making it unavailable for methylation, bioaccumulation, and biomagnification. It is noteworthy that the values found for the parameters NO_3_^−^, SO_4_^2−^, and Cl^−^, indicate contamination of the water body by the discharge of sanitary sewage and fertilizers. It is also noted that, according to the geological characteristics of the study area, there are no local geology contributions to these parameters. This indicates that diffuse sources are the main form of pollution for the study area. However, we anticipate that human impact through agricultural activities and urbanization presents some future challenges to the quality of water bodies in the DREPA. These results are valuable for establishing appropriate public policies and practices related to the management of public water supply reservoirs, and to verify the effectiveness of reducing anthropogenic mercury (Hg) emissions, as addressed in the 2017 Minamata convention.

## Figures and Tables

**Figure 1 ijerph-17-00154-f001:**
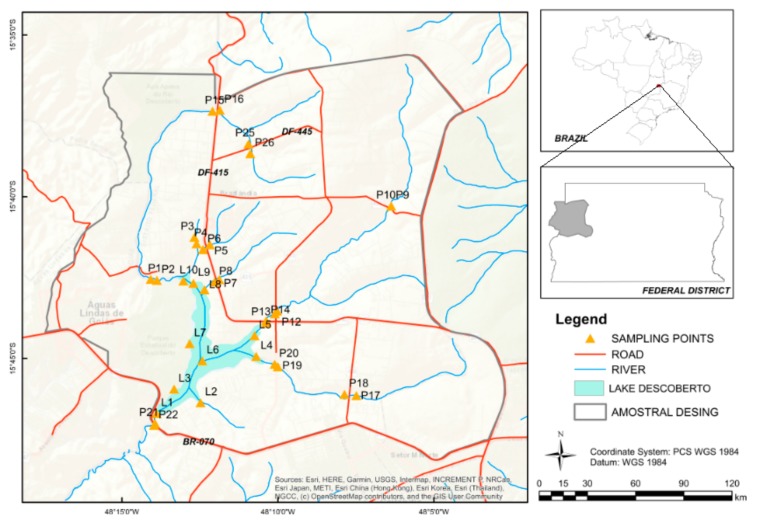
Map of the study area with projected coordinates of the sampling locations.

**Figure 2 ijerph-17-00154-f002:**
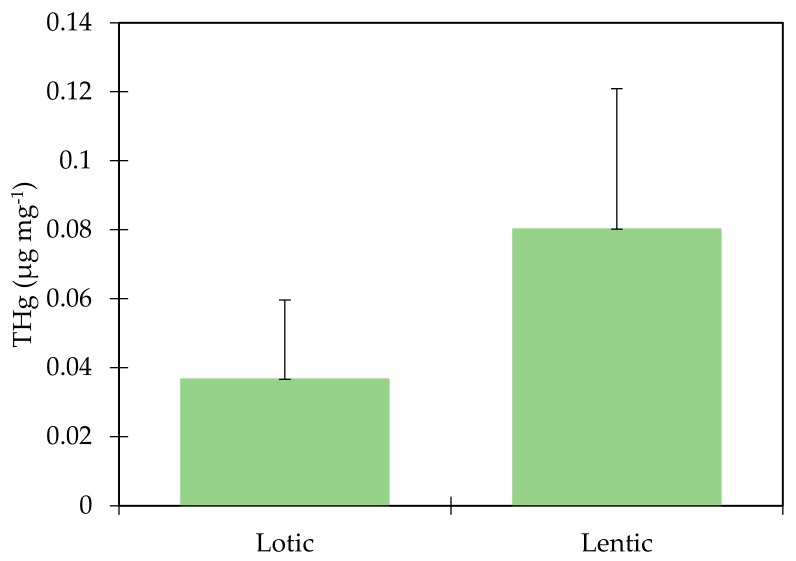
Comparison of THg concentration between the lotic environment and lentic environment U-test (*p* = 0.002).

**Figure 3 ijerph-17-00154-f003:**
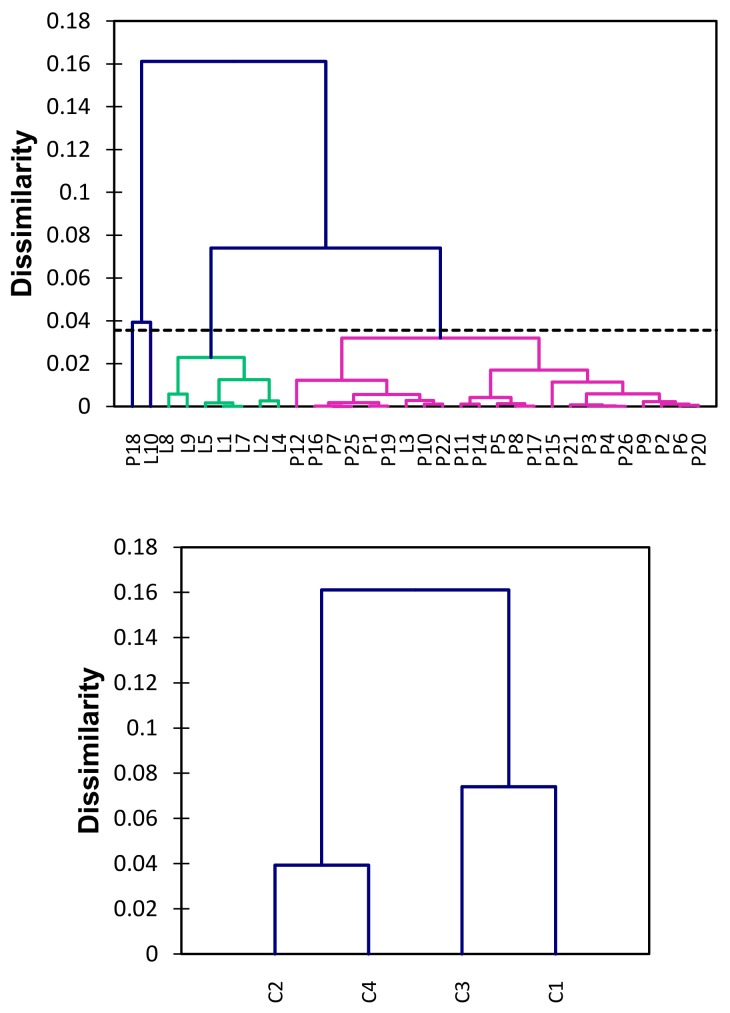
Dendrogram illustrating the hierarchical clustering of sediment samples grouped for THg concentration, ccc = 0.934.

**Figure 4 ijerph-17-00154-f004:**
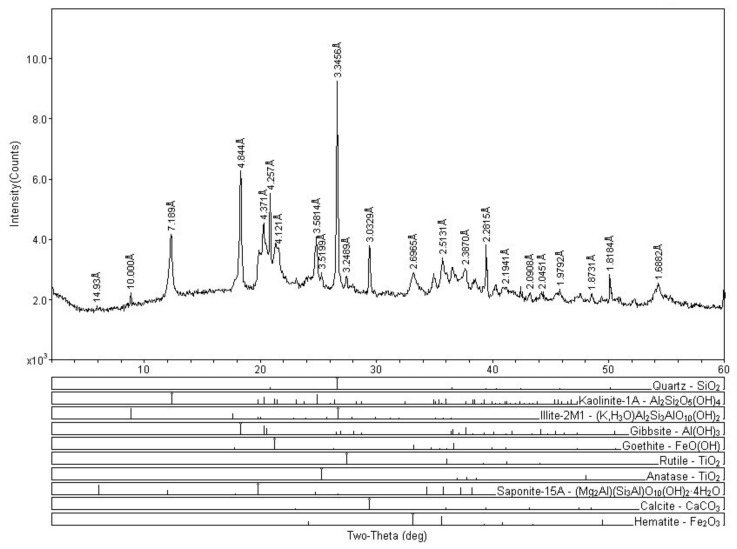
X-ray diffractogram of sediment sample (P15).

**Table 1 ijerph-17-00154-t001:** Summary of the variables analyzed according to the classification of the type of environment lotic (tributaries) and lentic (reservoir).

Variables	Lotic	Lentic	*p*
	Mean ± sd	Mean ± sd	
N	23	9	
*pH*	6.83 ± 0.54	6.81 ± 0.54	0.983
*EC* (µS cm^−1^*)*	30.65 ± 35.38	34.66 ± 16.37	0.116
**DO (mg L^−1^)**	26.35 ± 8.71	15.16 ± 5.72	0.001
*Tur (NTU)*	23.93 ± 14.62	22.93 ± 8.93	0.530
**T (°C)**	19.22 ± 1.60	22.30 ± 1.30	<0.001
*TDS* (mg L^−1^*)*	19.87 ± 22.92	22.55 ± 10.46	0.094
*HCO_3_^−^* (mg L^−1^*)*	8.97 ± 8.75	7.76 ± 1.34	0.516
*Cl^−^* (mg L^−1^*)*	1.31 ± 0.85	2.28 ± 2.69	0.049
*NO_3_^−^* (mg L^−1^*)*	0.80 ± 1.44	0.63 ± 0.16	0.020
**SO_4_^2−^ (mg L^−1^)**	0.46 ± 0.22	0.81 ± 0.18	<0.001
*THg* (µg g^−1^*)*	0.03 ± 0.02	0.08 ± 0.04	0.002
*VM (%)*	22.62 ± 18.46	20.51 ± 8.54	0.615

Sd: Standard deviation; *p* < 0.05: statistically significant; variables in bold: T-test for parametric variables; variables in italics: U-test for nonparametric variables.

**Table 2 ijerph-17-00154-t002:** Total mercury concentration in water bodies from other regions of Brazil.

Region	River	THg (µg g^−1^)	Aquatic System
GO	Tocantins River ^1^	0.04 ± 0.01	Lentic
RO	Madeira River ^1^	0.08 ± 0.01	Lentic
AM	Negro River ^1^	0.08 ± 0.02	Lotic
RO	Samuel Reservoir ^2^	0.04 ± 0.07	Lentic
DF	Descoberto Reservoir	0.08 ± 0.04	Lentic

^1^ [29]; ^2^ [17].

**Table 3 ijerph-17-00154-t003:** Descriptive statistics of sediment THg concentration in the study area, sediment reference values, and comparison of background values of this project with other studies.

Variables	Mean	Min	Max	ERL	ERM	RP	CDA	USA	DREPA
Hg (µg g^−1^)	0.05	0.01	0.18	0.15	1.3	0.1	0.05	0.05	0.04
Al (%)	12.2	7.41	16.7	−	−	5	6.8	5.5	11.4

ERL: effects range low, ERM: effects range median [21]; RP: Paracatu River [32]; CDA: Lake Coeur d’Alene [33]; USA [33].

**Table 4 ijerph-17-00154-t004:** Classification of the points collected in the Descoberto river basin environmental protection area, according to hierarchical grouping based on the THg median value.

Samples	Aquatic Environment	Road	Water Body	Class
P1	Lotic	Estrada	RD ^j^	1
P2	Lotic	Estrada	RD ^m^	1
P3	Lotic	Estrada	CCb ^m^	1
P4	Lotic	Estrada	CCb ^j^	1
P5	Lotic	BR080	CCp ^m^	1
P6	Lotic	BR080	CCp ^j^	1
P7	Lotic	BR080	CO ^m^	1
P8	Lotic	BR080	CO ^j^	1
P9	Lotic	DF430	RR ^j^	1
P10	Lotic	DF430	RR ^m^	1
P11	Lotic	DF445	RR ^j^	1
P12	Lotic	DF445	RR ^m^	1
P14	Lotic	BR251	RR ^m^	1
P15	Lotic	BR080	RD ^j^	1
P16	Lotic	BR080	RD ^m^	1
P17	Lotic	DF450	RP ^m^	1
P19	Lotic	DF180	RP ^m^	1
P20	Lotic	DF180	RP ^m^	1
P21	Lotic	BR070	RD ^j^	1
P22	Lotic	BR070	RD ^j^	1
P25	Lotic	DF415	Cco ^j^	1
P26	Lotic	DF415	Cco ^j^	1
L3	Lentic	RSV	RSV	1
P18	Lotic	DF450	RP ^j^	2
L1	Lentic	RSV	RSV	3
L2	Lentic	RSV	RSV	3
L4	Lentic	RSV	RSV	3
L5	Lentic	RSV	RSV	3
L7	Lentic	RSV	RSV	3
L8	Lentic	RSV	RSV	3
L9	Lentic	RSV	RSV	3
L10	Lentic	RSV	RSV	4

RSV: Reservoir; j: downstream; m: upstream; RD: Descoberto river; RR: Rodeador river; RP: Pedras river; CCb: Stream Caboquinha; CCp: Stream Chapadinha; CO: Stream Olaria; Cco: Capão da Onça.

**Table 5 ijerph-17-00154-t005:** Mineral constituents in sediment samples.

		QZ	KLN	ILT	GBS	RT	GTH	ANT	SPN	CAL	DLM	HEM
Lentic	L1	M+	M	M	m	m	m	m	−	m	m	−
	L2	M+	M	M	m	m	−	m	m	−	−	−
	L3	M+	M	M	m	m	m	m	m	−	−	−
	L4	M+	M	M	m	m	m	m	m	−	−	−
	L5	M+	M	M	m	m	m	m	m	−	−	−
	L7	M+	M	M	m	m	m	m	m	−	−	−
	L8	M+	M	M	m	m	m	m	m	−	−	−
	L9	M+	M	M	m	m	m	m	m	−	−	−
	L10	M+	M	M	m	m	−	m	m	−	−	−
lotic	P1	M+	M	M	m	m	m	m	m	−	−	−
	P3	M+	M	M	m	m	m	m	m	−	−	−
	P5	M+	M	M	m	m	m	m	m	−	−	−
	P7	M+	M	M	m	m	m	m	m	−	−	−
	P9	M+	M	M	m	m	−	m	m	−	−	−
	P11	M+	M	M	m	m	m	m	−	−	−	−
	P14	M+	M	M	m	m	m	m	m	−	−	−
	P15	M+	M	M	m	m	m	m	m	m	−	m
	P17	M+	M	M	m	m	m	m	m	−	−	−
	P19	M+	M	M	m	m	m	m	m	−	−	−
	P21	M+	M	M	m	m	m	m	m	−	−	−
	P25	M+	M	−	m	m	−	m	m	−	−	−

QZ = quartz, KLN = kaolinite, ILT = illite, GBS = gibbsite, RT = rutile, GTH = goethite, ANT = anatase, SNP = saponite, CAL = calcite, DLM = dolomite, HEM = hematite, M+ = predominant major constituent, M = major, m = minor, (-) = absent.

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
