# Peer review of "Evaluation of Total Mercury in Sediments of the Descoberto River Environmental Protection Area—Brazil"

_ijerph, 2019, doi:10.3390/ijerph17010154_

Round 1

Reviewer 1 Report

Comments are in text

Author Response

Comment: We thank the reviewer for the constructive criticism.

Point 1: (Abstract - Line 23) reduce the number of decimals in p values.

Response 1: Done.

Point 2: (Abstract - Line 24) THg concentrations showed significant difference in sediments or in water?

Response 2: In sediments.

Point 3: (Materials and Methods - Line 127) highlighted sentence: “with 10% nitric acid solution”.

Response 3: previously cleaned in acidic solution (HCl - 10%) and left in solution for at least 8h, rinsed with water from the Milli-Q purification system.

Point 4: (Materials and Methods - Line 131) “Please, describe details”.

Response 4: As suggested by the reviewer, we reformulated the description in more detail.

Point 5: (Materials and Methods - Lines 142 - 143) “Please, describe parameters etc.”

Response 5: As suggested by the reviewer, we reformulated the description of the parameters in more detail.

Point 6: (Materials and Methods - Line 162): delete space.

Response 6: Done.

Point 7: (Materials and Methods - Line 165) highlighted sentence “Statistical analysis”.

Response 7: we reformulated the description of the statistical analysis in more detail.

Point 8: (Results and Discussion - Line 204) insert space.

Response 8: Done.

Point 9: (Results and Discussion - Line 207) reduce number of significant digits.

Response 9: Done.

Point 10: (Results and Discussion - Line 209) English edition.

Response 10: Done.

Point 11: (Results and Discussion - Lines 212-213) correct superscript “L-1”.

Response 11: Done.

Point 12: (Results and Discussion - table 1, lines 224 -225) insert space and “there is no statistical analysis”.

Response 12: Done.

Comment: Description of the statistical analysis has been reformulated in the materials and methods section, and the analysis improved in the results and discussion section.

Point 13: (Results and Discussion - Line 236) English edition and reduce number of significant digits.

Response 13: Done.

Point 14: (Results and Discussion – Figure 2, lines 249 - 251) insert unit; English edition and reduce number of significant digits.

Response 14: Done.

Point 15: (Results and Discussion - Line 271) Hg2+, upperscript.

Response 15: Done.

Point 16: (Results and Discussion - Lines 293-295) different units: Hg = µg g-1 and Al = %?

Response 16: Yes, just to express the result, due to the amount of the element in the sample. But to calculate the geochemical indices the same units of measurement were used for the two elements: Al and Hg = µg g-1.

Point 17: (Results and Discussion – Figure 3, line 338) which one is lentic and which one is lotic? it should be described.

Response 17: Caption has been reworked: Dendrogram illustrating the hierarchical clustering of sediment samples grouped for THg concentration, ccc = 0.934.

Point 18: (References - Line 471) delete space.

Response 18: Done.                          

Point 19: (References - Line 480) delete space.

Response 19: Done.

Point 20: (References - Line 484) delete space.

Response 20: Done.

Point 21: (References - Line 485) delete space.

Response 21: Done.

Reviewer 2 Report

In this study, the authors investigated the distribution of total mercury in sediments of the Descoberto River Environmental Protection Area, Brazil. And they evaluated the environmental threats and the anthropogenic enrichment of mercury in this area. As mentioned by the authors in the Introduction, studies that consider the local variables of each regions may be necessary for contributing to land use management and avoiding the deterioration of aquatic ecosystems. Moreover, it will be a fact that evaluation of mercury levels in tropical reservoirs without direct influence from known anthropogenic sources has not been frequently conducted. In this study, however, there is little new and original contribution, and also the methodology used is very commonplace. Hence, I think that this topic is better suited for local journal rather than international journal. Other minor comments are given below.

Lines 28–29: Whether most of the samples are moderately polluted through atmospheric deposition due to vehicular traffic and agriculture is not discussed in the text.

Line 126: What types of water samples (bottom water or overlying water on sediment; depth in water column?) were collected? The authors should also explain how to collect the water samples.

Lines 141–142: What is the recovery efficiency of mercury measured using a standard reference material (NIST 2709)?

Lines 182–188: The authors should explain how to measure Al concentration in the sediment samples in this study.

Table 1, Figure 2, and Figure 3: Revise decimal point “,” to “.”

Table 3: What are abbreviations of RP and APARD? The number of significant figures of the Hg and Al concentrations is too many, three at most.

Lines 235–238: Did the authors measure the particle sizes of the sediment samples? The difference in mercury concentration in the sediments between lentic and lotic environments may depend on the difference in the particle sizes of sediments between both environments, because mercury is generally enriched in fine particles.

Reviewer 3 Report

This is an interesting paper, addressing an important topic. However, there are some shortcomings in data analysis and results presentation, that must be solved prior to publication. Also some English editing is required. 

Round 2

Reviewer 2 Report

I think that the revised manuscript has been successfully improved. Hence, I recommend the publication of this paper in the present form.